# National survey of attitudes towards and intentions to vaccinate against COVID-19: implications for communications

Martine Stead,[1] Curtis Jessop,[2] Kathryn Angus [ID],[1] Helen Bedford,[3] Michael Ussher,[1,4] Allison Ford,[1] Douglas Eadie,[1] Andy MacGregor,[2] Kate Hunt [ID],[1] Anne Marie MacKintosh[1]

¹Institute for Social Marketing and Health, University of Stirling, Stirling, UK
²NatCen The National Centre for Social Research, London, UK
³Great Ormond Street Institute of Child Health, University College London, London, UK
⁴Population Health Research Institute, St George's University of London, London, UK

**Correspondence to**
Martine Stead;
martine.stead@stir.ac.uk

## ABSTRACT

**Objectives** To examine public views on COVID-19 vaccination and consider the implications for communications and targeted support.

**Design** Cross-sectional study.

**Setting** Online and telephone nationally representative survey in Great Britain, January to February 2021.

**Participants** 4978 adults. Survey response rate was 84%, among the 5931 panellists invited.

**Main outcome measures** Sociodemographic characteristics (age, gender, ethnicity, education, financial status), COVID-19 status, vaccine acceptance, trust in COVID-19 vaccination information sources, perceptions of vaccination priority groups and perceptions of importance of second dose.

**Results** COVID-19 vaccine acceptance (83%) was associated with increasing age, higher level of education and having been invited for vaccination. Acceptance decreased with unconfirmed past COVID-19, greater financial hardship and non-white British ethnicity; black/black British participants had lowest acceptance. Overall, healthcare and scientific sources of information were most trusted. Compared with white British participants, other ethnicities had lower trust in healthcare and scientific sources. Those with lower educational attainment or financial hardship had lower trust in healthcare and scientific sources. Those with no qualifications had higher trust in media and family/friends. While trust was low overall in community or faith leaders, it was higher among those with Asian/Asian British and black/black British ethnicity compared with white British participants. Views of vaccine prioritisation were mostly consistent with UK official policy but there was support for prioritising additional groups. There was high support for having the second vaccine dose.

**Conclusions** Targeted engagement is needed to address COVID-19 vaccine hesitancy in non-white British ethnic groups, in younger adults, and among those with lower education, greater financial hardship and unconfirmed past infection. Healthcare professionals and scientific advisors should play a central role in communications and tailored messaging is needed for hesitant groups. Careful communication around vaccination prioritisation continues to be required.

### STRENGTHS AND LIMITATIONS OF THIS STUDY

⇒ The survey was conducted at the start of vaccine roll-out giving timely insight into COVID-19 vaccine acceptance/hesitancy and trusted information sources when individuals' decision making was real rather than hypothetical.

⇒ Results come from a large probability-based sample, representative of adults in Great Britain, which was sufficiently large to examine ethnicity in detail.

⇒ The survey did not include those who are institutionalised (eg, prisoners), notably difficult to reach populations (eg, homeless) or those not speaking English (therefore, our ethnic minority sample may under-represent certain views).

⇒ The survey benefited from a rigorous design, with questionnaire development informed by cognitive interviews conducted with a broad range of individuals.

⇒ A cross-sectional survey cannot infer causality; although variables likely to be important in vaccine acceptance were included, the results are exploratory.

## INTRODUCTION

Widespread vaccination is likely to be one of the most effective ways of controlling the COVID-19 pandemic, and is central to the UK government's recovery strategy. The UK vaccine programme began in December 2020, prioritising older adults in care homes and their carers, those aged over 80, and front-line health and social care workers.[1] Administration of first doses of vaccination to the adult population, by decade of age, is to be completed by July 2021. Uncertainty or unwillingness to accept vaccination—'vaccine hesitancy'[2]—threatens comprehensive vaccination.[3 4] Before the introduction of a COVID-19 vaccine, UK surveys reported that 64%—82% of adults were willing to be vaccinated.[5–12] Most of these studies used non-probability samples, introducing selection

bias and limiting generalisability. Increased vaccine confidence has been reported since vaccination commenced[13]; possibly due to increased COVID-19 cases and deaths, a further UK lockdown in early 2021, and, increasingly, vaccination becoming the social norm. It is important to examine vaccine acceptance when people are making active, rather than hypothetical, decisions about vaccination. This also provides insight into potential acceptance of repeat COVID-19 vaccination and boosters.[14]

UK uptake has been high (94% of adults surveyed in April reported uptake or intention to accept vaccination),[13] but there remain concerns about uptake in subpopulations, such as younger adults and some ethnic minorities,[15] giving rise to initiatives such as social media campaigns featuring non-white celebrities.[16] Robust, timely data are needed to identify the characteristics of groups with lower acceptance and the information sources they trust, to inform targeted interventions. It is also important to assess whether attitudes towards COVID-19 vaccination have been affected by specific events and media coverage. Two issues in the UK merit particular attention. First, the government followed recommendations to offer the vaccine to priority groups.[1] If this approach is continued, it is important to examine its acceptability and any implications for communications. Second, the government decided, on 30 December 2020, to deviate from recommended protocols for the Pfizer-BioNTech vaccine by extending the interval between doses to up to 12 weeks[1]; this precipitated concerns that it may lead to reduced willingness to be vaccinated or to have a second dose.[17]

We conducted a survey in early 2021, using probability sampling, to examine public views on COVID-19 vaccination and consider the implications for communications. During this period, most people aged over 80 had been invited to have a vaccine and invitations were being extended to those aged over 70, with other age groups advised they would be invited in the coming months.

## METHODS

We administered a cross-sectional survey with adults (aged 18+) in Great Britain (GB) in January and February 2021. This paper follows the STROBE Statement (STrengthening the Reporting of OBservational studies in Epidemiology) for reporting cross-sectional studies.[18]

### Questionnaire development and testing

The questionnaire was informed by a review of studies on public attitudes towards and experiences of vaccines and COVID-19. Existing measures were adapted[5 19 20] and new questions developed.

The questionnaire was cognitively tested with members of the public to ensure understandability.[21] Interviews were conducted with 20 individuals recruited by an external fieldwork agency. A purposive sampling approach was employed, with quotas used to ensure people with a mix of genders, ages, parental status, likelihood of accepting a COVID-19 vaccination and experiences of shielding were recruited. The questionnaire was subsequently revised based on these interviews. Final revisions reflected changes in the UK's vaccine roll-out. The questionnaire covered: vaccine acceptance, trust in vaccine information sources, perception of priority groups, COVID-19 status and perceived importance of a second dose. The questionnaire is provided in online supplemental material, methods S1.

### Sample and data collection

The target population for the study was adults (18+) living in GB. The survey was administered to the probability-based NatCen Panel,[22] recruited from the 2018, 2019 and 2020 waves of the British Social Attitudes survey (BSA), with participants randomly selected from England, Wales and Scotland. All BSA respondents who agreed to join the panel, had not requested to leave or become inactive were invited to take part, maintaining the random probability design. Data were collected through online and telephone interviews (conducted 14 January 2021 to 7 February 2021). Panellists were sent reminders and offered a small financial sum (£5–£20 depending on interview duration and whether participant had characteristics which are typically under-represented in survey samples) in recognition of their contribution. Participants who did not initially take part online, and for whom a telephone number was available, were followed up by a telephone interviewer and encouraged to take part online or given the opportunity to take part on the telephone. Among 5931 panellists invited, the survey response rate was 84%, with 4978 completing it (4776 online, 202 by telephone). Online supplemental table S1 details overall response rate, accounting for non-response at the panel recruitment stage and panel attrition. Data were weighted for non-response and to be representative of the GB adult population (see online supplemental material, methods S2).

### Measures

#### Sociodemographic and other characteristics

Data on age, gender, ethnicity, education, country, urban/rural status and financial status were obtained from existing information on NatCen panellists. Full details of subgroups of each variable are provided in tables 1 and 2. Age was categorised into bands from 18 to 29 years then 10-year bands up to 80+. Self-assigned ethnicity was recorded in six categories, and education in five categories according to highest qualification. As indices of multiple deprivation were not available, self-reported financial status was used. COVID-19 status was derived from two items: (1) 'Have you been officially diagnosed with the coronavirus (COVID-19)?' (yes/no/don't know); those answering other than 'yes' were asked: (2) 'Do you think you have ever had the coronavirus (COVID-19)?' (yes-definitely/yes-probably/no-probably not/no-definitely not/don't know).

#### Vaccine measures

Vaccine acceptance was derived from five items: (1) 'Have you been offered a vaccine for COVID-19?' (yes/

**Table 1** Sample characteristics

| | Unweighted | | Weighted | |
|---|---|---|---|---|
| | n | % | n | % |
| Age | | | | |
| 18–29 | 464 | 9.4 | 824 | 16.7 |
| 30–39 | 772 | 15.6 | 852 | 17.3 |
| 40–49 | 848 | 17.1 | 806 | 16.3 |
| 50–59 | 904 | 18.3 | 867 | 17.6 |
| 60–69 | 1011 | 20.4 | 711 | 14.4 |
| 70–79 | 773 | 15.6 | 657 | 13.3 |
| 80+ | 178 | 3.6 | 218 | 4.4 |
| Gender | | | | |
| Male | 2136 | 42.9 | 2402 | 48.3 |
| Female | 2830 | 56.9 | 2567 | 51.6 |
| Other | 10 | 0.2 | 7 | 0.1 |
| Ethnicity | | | | |
| White British | 4261 | 86.3 | 3999 | 81.2 |
| Any other white background | 319 | 6.5 | 335 | 6.8 |
| Mixed or multiple ethnic groups | 64 | 1.3 | 100 | 2.0 |
| Asian or Asian British | 164 | 3.3 | 306 | 6.2 |
| Black or black British | 67 | 1.4 | 101 | 2.1 |
| Other | 62 | 1.3 | 81 | 1.6 |
| Country | | | | |
| England | 4369 | 87.9 | 4291 | 86.3 |
| Scotland | 390 | 7.8 | 442 | 8.9 |
| Wales | 212 | 4.3 | 237 | 4.8 |
| Urban/rural status* | | | | |
| Urban | 3789 | 76.2 | 4006 | 80.6 |
| Rural | 1182 | 23.8 | 965 | 19.4 |
| Highest educational qualification | | | | |
| Degree or equivalent, and above | 2503 | 50.4 | 2077 | 41.8 |
| A levels or vocational level 3 or equivalent and above, but below degree | 1005 | 20.2 | 1131 | 22.8 |
| Other qualifications below A levels or vocational level 3 or equivalent | 788 | 15.9 | 838 | 16.9 |
| Other qualification | 256 | 5.2 | 304 | 6.1 |
| No qualifications | 416 | 8.4 | 618 | 12.4 |
| Subjective financial status | | | | |
| Living comfortably | 1552 | 31.2 | 1289 | 26.0 |
| Doing alright | 2028 | 40.8 | 2035 | 40.9 |
| Just about getting by | 975 | 19.6 | 1132 | 22.8 |
| Finding it quite difficult | 271 | 5.5 | 337 | 6.8 |
| Finding it very difficult | 142 | 2.9 | 175 | 3.5 |
| COVID-19 status | | | | |
| Diagnosed with COVID-19 | 241 | 4.8 | 294 | 5.9 |
| Think definitely had COVID-19 | 140 | 2.8 | 172 | 3.5 |
| Think probably had COVID-19 | 710 | 14.3 | 755 | 15.2 |

Continued

**Table 1** Continued

| | Unweighted | | Weighted | |
|---|---|---|---|---|
| | n | % | n | % |
| Think probably not had COVID-19 | 1945 | 39.1 | 1880 | 37.8 |
| Think definitely not had COVID-19 | 1393 | 28.0 | 1305 | 26.2 |
| Don't know if had COVID-19 | 547 | 11.0 | 566 | 11.4 |

*England and Wales, based on Office for National Statistics definition of urban as population greater than 10 000. Scotland based on Scottish Government definition of urban as population greater than 3000.

no). Those answering 'yes' were asked: (2) 'And have you had that vaccine?' (yes/no). Participants who had been offered but not yet had the vaccine were then asked: (3) 'And do you intend to have that vaccine?' (yes/no/not sure). Participants who had not yet been offered the vaccine were asked: (4) 'Would you accept the vaccine for yourself if it is offered to you?' (yes/no/not sure). Those answering 'not sure' were asked: (5) 'If you had to choose, if a COVID-19 vaccine became publicly available and you were offered it, would you accept the vaccine for yourself?' (yes/no/I'm really not sure). Participants were classed as: 'Accepted/accepting' if they answered 'yes' to any of items 2, 3, 4 or 5; 'Uncertain' if they answered 'not sure' to item 3 or 'I'm really not sure' to item 5; and 'Refused/refusing' if they answered 'no' to items 3, 4 or 5.

Trust in information sources was assessed for 13 sources: 'To what extent, if at all, would you trust information about a COVID-19 vaccine from each of the following sources?' (see table 3): completely (1); a great deal (2); somewhat (3); very little (4); not at all (5).

Perceptions of vaccine priority groups were assessed across 11 groups (see table 4): 'Below are some groups that some people say should be the first to be offered a COVID-19 vaccine. For each one, how high a priority do you think it is that they get a COVID-19 vaccine, or do you not think they should be offered the vaccine at all?': 1 'One of the first', 5 'One of the last', with an additional option 'They should not be offered a vaccine'.

Perceived importance of receiving the second dose of the vaccine was assessed with: 'How important, if at all, do you think it is for people to get the second injection of the COVID-19 vaccine?': very important (1); fairly important (2); not very important (3); not at all important (4).

### Data analysis
Descriptive data, including bivariate analyses, were weighted to be representative of British adult population. Initial bivariate analyses, using $\chi^2$ tests, examined correlates of vaccine acceptance and trust in sources of information about COVID-19 vaccination. Multivariate logistic regression was conducted to examine differences in vaccine acceptance controlling for sociodemographic variables, vaccine offer and COVID-19 status. The dependent variable dichotomised those classed as accepted/intend to accept vs uncertain/refused/intend to refuse.

Age was entered as a categorical variable and the 'difference' contrast within SPSS logistic regression was used to test influence of each increasing age group, relative to younger ages (eg, 30–39 vs 18–29; 80+ vs 18–79) (see table 2). Sociodemographic variation in trust in information sources was examined using multivariate logistic regressions. For each information source, the dependent variable dichotomised the 5-point scale into trusting completely or a great deal vs somewhat/very little/not at all. Cases were excluded from the logistic regressions if they had missing data on the dependent or any independent variables. All logistic regressions were conducted on unweighted data as sociodemographic variables were included as control variables. For each information source, logistic regression analysis examined likelihood of trust (completely/a great deal v somewhat/very little/not at all) by sociodemographic characteristics (online supplemental tables S2–S14). Given the large sample size in this study, the threshold for statistical significance was set at $p<0.01$. Data were analysed using SPSS V.27.

### Public and patient involvement
The questionnaire was cognitively tested by members of the public to ensure understandability (see the section 'Questionnaire development and testing' above).

## RESULTS
### Sample characteristics
The weighted sample comprised adults aged 18 and over (see table 1). Over half (52%) were female and 81% were white British. Around two-thirds reported 'living comfortably'/'doing alright', while one in ten rated their financial status as 'quite' or 'very difficult'. Just over two-fifths were educated to degree level or above, while for almost a quarter their highest qualification was A level or equivalent. A minority (12%) had no qualifications. A minority indicated having been diagnosed with COVID-19 (6%); nearly two-thirds thought they probably or definitely had not had COVID-19; 11% were unsure.

### Vaccine offer and acceptance
At the time of the survey, 14% (n=716) had been offered the vaccine. Of these, 92% (n=658) had accepted or intended to, 4% (n=29) were uncertain and 4% (n=29) had refused or intended to refuse.

**Table 2** Association between vaccine acceptance and sociodemographic variables—(A) bivariate results and (B) multivariate logistic regression.

| | (A) Bivariate associations between vaccine acceptance and sociodemographics % Accepted/Intend to accept (weighted) $\chi^2$ test for differences by demographics | | | | (B) Logistic regression of vaccine acceptance 1=Accepted/Intend to accept (4294), 0=uncertain/refused/intend to refuse (600) | | | | |
| | n | % | $\chi^2$ (df) | P value | N | AOR* | 95% CI lower | 95% CI upper | P value |
|---|---|---|---|---|---|---|---|---|---|
| Gender | | | 2.154 (2) | 0.341 | | | | | 0.085 |
| Male | 2012 | 83.8 | | | 2097 | ref | | | |
| Female | 2117 | 82.5 | | | 2788 | 0.82 | 0.67 | 0.99 | 0.036 |
| Other | 5 | 71.4 | | | 9 | 0.47 | 0.09 | 2.45 | 0.369 |
| Age | | | 274.733 (6) | <0.001 | | | | | <0.001 |
| 18–29 | 613 | 74.4 | | | 459 | ref | | | |
| 30–39 vs 18–29 | 618 | 72.5 | | | 761 | 0.89 | 0.66 | 1.20 | 0.448 |
| 40–49 vs 18–39 | 640 | 79.3 | | | 835 | 1.43 | 1.12 | 1.83 | 0.004 |
| 50–59 vs 18–49 | 745 | 85.9 | | | 896 | 1.92 | 1.49 | 2.46 | <0.001 |
| 60–69 vs 18–59 | 659 | 92.7 | | | 1003 | 3.21 | 2.37 | 4.34 | <0.001 |
| 70–79 vs 18–69 | 629 | 95.7 | | | 763 | 3.31 | 2.22 | 4.95 | <0.001 |
| 80+ vs 18–79 | 209 | 95.9 | | | 177 | 2.19 | 0.92 | 5.21 | 0.078 |
| Education/highest qualification | | | 56.056 (4) | <0.001 | | | | | <0.001 |
| No qualifications | 495 | 80.1 | | | 411 | ref | | | |
| Degree or equivalent and above | 1811 | 87.2 | | | 2454 | 3.03 | 2.17 | 4.23 | <0.001 |
| A levels/vocational level 3 or equivalent | 909 | 80.4 | | | 990 | 1.80 | 1.27 | 2.55 | <0.001 |
| Other qual'ns below A level/voc level 3 | 694 | 82.7 | | | 784 | 1.50 | 1.05 | 2.15 | 0.026 |
| Other qualification | 223 | 73.4 | | | 255 | 0.90 | 0.58 | 1.39 | 0.632 |
| Financial status | | | 168.660 (4) | <0.001 | | | | | <0.001 |
| Living comfortably | 1162 | 90.1 | | | 1533 | ref | | | |
| Doing alright | 1749 | 86.0 | | | 1998 | 0.89 | 0.69 | 1.15 | 0.383 |
| Just about getting by | 848 | 74.9 | | | 959 | 0.52 | 0.39 | 0.69 | <0.001 |
| Finding it quite difficult | 261 | 77.2 | | | 266 | 0.74 | 0.50 | 1.10 | 0.139 |
| Finding it very difficult | 111 | 63.4 | | | 138 | 0.35 | 0.22 | 0.55 | <0.001 |
| Country | | | 3.171 (2) | 0.205 | | | | | 0.326 |
| England | 3581 | 83.5 | | | 4302 | ref | | | |
| Scotland | 356 | 80.5 | | | 384 | 0.82 | 0.59 | 1.13 | 0.220 |
| Wales | 192 | 81.0 | | | 208 | 0.80 | 0.51 | 1.26 | 0.345 |
| Urban/rural | | | 34.517 (1) | <0.001 | | | | | |
| Urban | 3266 | 81.5 | | | 3729 | ref | | | |
| Rural | 863 | 89.4 | | | 1165 | 1.28 | 1.00 | 1.65 | 0.051 |
| Ethnicity | | | 246.434 (5) | <0.001 | | | | | <0.001 |
| White British | 3482 | 87.1 | | | 4226 | ref | | | |
| Any other white background | 254 | 75.8 | | | 318 | 0.55 | 0.40 | 0.76 | <0.001 |
| Mixed or multiple ethnic groups | 62 | 61.4 | | | 62 | 0.39 | 0.21 | 0.71 | 0.002 |

**Table 2** Continued

| | (A) Bivariate associations between vaccine acceptance and sociodemographics % Accepted/Intend to accept (weighted) $\chi^2$ test for differences by demographics | | | | (B) Logistic regression of vaccine acceptance 1=Accepted/Intend to accept (4294), 0=uncertain/refused/intend to refuse (600) | | | | |
| | n | % | $\chi^2$ (df) | P value | N | AOR* | 95% CI lower | 95% CI upper | P value |
|---|---|---|---|---|---|---|---|---|---|
| Asian or Asian British | 188 | 61.4 | | | 161 | 0.41 | 0.28 | 0.61 | <0.001 |
| Black or black British | 59 | 58.4 | | | 67 | 0.25 | 0.14 | 0.43 | <0.001 |
| Other | 59 | 72.8 | | | 60 | 0.42 | 0.23 | 0.79 | 0.007 |
| Whether been offered vaccine | | | 45.924 (1) | <0.001 | | | | | |
| No | 3479 | 81.6 | | | 4227 | ref | | | |
| Yes | 658 | 91.9 | | | 667 | 1.73 | 1.24 | 2.43 | 0.001 |
| COVID-19 status | | | 72.865 (4) | <0.001 | | | | | <0.001 |
| Think probably or definitely <u>not</u> had COVID-19 | 2741 | 86.1 | | | 3288 | ref | | | |
| Diagnosed with COVID-19 | 218 | 74.4 | | | 240 | 0.89 | 0.60 | 1.33 | 0.575 |
| Think definitely had COVID-19 | 118 | 68.2 | | | 140 | 0.40 | 0.26 | 0.60 | <0.001 |
| Think probably had COVID-19 | 598 | 79.1 | | | 691 | 0.71 | 0.56 | 0.91 | 0.006 |
| Don't Know if had COVID-19 | 462 | 81.5 | | | 535 | 0.73 | 0.55 | 0.97 | 0.031 |
| | | | | | Hosmer & Lemeshow $\chi^2$=7.444, df=8, p=0.490. | | | | |
| | | | | | Final model $\chi^2$=497.429, df=29, p<0.001 | | | | |
| | | | | | Nagelkerke=0.184 | | | | |
| | | | | | Cases correctly classified: 88.1%. | | | | |
| | | | | | 84 cases excluded due to missing data on one or more independent variables. | | | | |

*Adjusted for all other variables in the model.
AOR, adjusted OR; 95% CI, 95% confidence interval; ref, reference category.

Among those not yet offered the vaccine, 82% (n=3479) intended to accept, while 11% (n=471) were uncertain and 7% (n=311) indicated they would refuse. Overall, the acceptance level was 83% (n=4137), with 10% (n=502) uncertain and 7% (n=340) refusing.

Multivariate logistic regression, with vaccine acceptance as the outcome variable (accepted/accepting vs refused/refusing/uncertain), indicated likelihood of acceptance increased with age (table 2). For example, those aged 40–49 were more likely than 18–39 years to indicate acceptance (adjusted OR, AOR=1.43, 95% CI (1.12 to 1.83, p=0.004) as were 70–79 years compared with 18–69 years (AOR=3.31, 95% CI (2.22 to 4.95), p<0.001). Acceptance was also positively associated with education. Those with at least a degree were three times as likely to indicate acceptance (AOR=3.03, 95% CI (2.17 to 4.23), p<0.001) and those educated to A level or equivalent were

nearly twice as likely (AOR=1.80, 95% CI (1.27 to 2.55), p<0.001), compared with people without qualifications. Lower acceptance was also associated with financial hardship and ethnicity. For example, compared with those 'living comfortably', people 'finding it very difficult' were much less likely to accept the vaccine (AOR=0.35, 95% CI (0.22 to 0.55), p<0.001). Compared with white British participants, those from other ethnic groups were less likely to accept the vaccine. Black/black British participants had the lowest likelihood of accepting (AOR=0.25, 95% CI (0.14 to 0.43), p<0.001). This is illustrated in the descriptive data too, with 87% of white British participants indicating vaccine acceptance compared with 58% among black/black British, 61% among mixed/multiple ethnic groups and 61% among Asian/Asian British.

After controlling for demographic variables, vaccine acceptance was positively associated with having been

**Table 3** Trust in potential sources of information on COVID-19 vaccine

| | Level of trust (trust completely(1)…not at all (5)) | | | | | | | | | | |
| | Completely (1) | | A great deal (2) | | Somewhat (3) | | Very little (4) | | Not at all (5) | | | |
| Source: | n | % | n | % | n | % | n | % | n | % | Mean | SD |
|---|---|---|---|---|---|---|---|---|---|---|---|---|
| The NHS | 2084 | 41.9 | 1902 | 38.3 | 701 | 14.1 | 155 | 3.1 | 127 | 2.5 | 1.86 | 0.95 |
| Doctors, nurses or other healthcare professionals | 1918 | 38.6 | 2092 | 42.1 | 714 | 14.4 | 154 | 3.1 | 90 | 1.8 | 1.87 | 0.90 |
| Scientific and medical advisers | 1798 | 36.2 | 2101 | 42.3 | 792 | 15.9 | 160 | 3.2 | 121 | 2.4 | 1.94 | 0.93 |
| The World Health Organisation (WHO) | 1313 | 26.4 | 2016 | 40.6 | 1070 | 21.6 | 310 | 6.2 | 256 | 5.1 | 2.23 | 1.07 |
| Pharmacists | 999 | 20.1 | 1973 | 39.7 | 1434 | 28.8 | 341 | 6.9 | 226 | 4.5 | 2.36 | 1.02 |
| The UK government | 654 | 13.2 | 1542 | 31.1 | 1739 | 35.1 | 614 | 12.4 | 402 | 8.1 | 2.71 | 1.10 |
| The Scottish Government/ The Welsh Assembly* | 118 | 17.4 | 189 | 27.9 | 207 | 30.5 | 88 | 13.1 | 75 | 11.1 | 2.72 | 1.21 |
| Drug companies who manufacture vaccines | 406 | 8.2 | 1064 | 21.4 | 2065 | 41.6 | 771 | 15.5 | 661 | 13.3 | 3.04 | 1.11 |
| Family and friends | 343 | 6.9 | 876 | 17.6 | 2230 | 44.9 | 977 | 19.7 | 542 | 10.9 | 3.10 | 1.04 |
| The media (eg, newspapers, magazines, television, radio) | 86 | 1.7 | 302 | 6.1 | 1567 | 31.5 | 1433 | 28.9 | 1580 | 31.8 | 3.83 | 1.00 |
| Faith or community leaders | 131 | 2.6 | 124 | 2.5 | 619 | 12.5 | 827 | 16.7 | 3264 | 65.7 | 4.40 | 0.98 |
| Social media (eg, Twitter, Facebook, Instagram) | 65 | 1.3 | 69 | 1.4 | 506 | 10.2 | 1267 | 25.5 | 3056 | 61.6 | 4.45 | 0.83 |
| Celebrities and social media influencers | 60 | 1.2 | 71 | 1.4 | 493 | 9.9 | 1175 | 23.6 | 3170 | 63.8 | 4.47 | 0.82 |

Base: All participants (weighted). Missing cases range from n=3 to n=27. List order was randomised for each participant.
*Base: all participants in Scotland or Wales, n=679 (weighted).
NHS, National Health Service; SD, standard deviation.

invited for vaccination (AOR=1.73, 95% CI (1.24 to 2.43), p=0.001), but negatively associated with COVID-19 status. Compared with those who had 'probably not' or 'definitely not' had COVID-19, those who thought they had 'definitely' or 'probably' had COVID-19 were less likely to indicate acceptance (AOR=0.40, 95% CI (0.26 to 0.60), p<0.001 and AOR=0.71, 95% CI (0.56 to 0.91), p=0.006, respectively). Confirmed diagnosis with COVID-19 was not significantly associated with vaccine acceptance, after controlling for demographic variables.

### Trust in information sources

The three most trusted information sources were: the National Health Service (NHS); doctors/nurses/other healthcare professionals and scientific and medical advisers. These groups were trusted 'completely/a great deal' by around 80% of participants (table 3). Only 44% trusted the UK government 'completely/a great deal'. The three least trusted sources were celebrities and social media influencers, social media, and faith or community leaders; around two-thirds indicated they would have no

**Table 4** Views on priority groups for vaccination: who should be first and last groups vaccinated

| | Should not be offered | | Priority of being offered* | | | | | | | | | | | | |
| | | | One of the first (1) | | (2) | | (3) | | (4) | | One of the last (5) | | | |
| | n | % | n | % | n | % | n | % | n | % | n | % | Mean† | SD |
|---|---|---|---|---|---|---|---|---|---|---|---|---|---|---|
| Doctors, nurses and other healthcare professionals | 33 | 0.7 | 4472 | 90.0 | 280 | 5.6 | 83 | 1.7 | 15 | 0.3 | 83 | 1.7 | 1.17 | 0.63 |
| People with serious health conditions which mean they are vulnerable to COVID-19 | 35 | 0.7 | 4017 | 80.9 | 671 | 13.5 | 129 | 2.6 | 35 | 0.7 | 77 | 1.6 | 1.27 | 0.69 |
| Care home workers | 36 | 0.7 | 3926 | 79.0 | 683 | 13.8 | 197 | 4.0 | 58 | 1.2 | 66 | 1.3 | 1.31 | 0.72 |
| Residents in a care home | 47 | 0.9 | 3593 | 72.4 | 734 | 14.8 | 337 | 6.8 | 123 | 2.5 | 131 | 2.6 | 1.47 | 0.93 |
| People aged 80 or over | 49 | 1.0 | 3613 | 72.9 | 706 | 14.2 | 304 | 6.1 | 118 | 2.4 | 168 | 3.4 | 1.48 | 0.96 |
| Social care workers | 33 | 0.7 | 2683 | 54.0 | 1348 | 27.2 | 683 | 13.8 | 143 | 2.9 | 75 | 1.5 | 1.70 | 0.92 |
| Schoolteachers | 47 | 0.9 | 2098 | 42.2 | 1621 | 32.6 | 886 | 17.8 | 223 | 4.5 | 94 | 1.9 | 1.90 | 0.97 |
| People with jobs that involve direct contact with members of the public | 45 | 0.9 | 1864 | 37.5 | 1603 | 32.3 | 1157 | 23.3 | 228 | 4.6 | 70 | 1.4 | 1.99 | 0.96 |
| People aged 31–50 | 43 | 0.9 | 154 | 3.1 | 614 | 12.4 | 2096 | 42.2 | 1486 | 30.0 | 568 | 11.4 | 3.35 | 0.95 |
| People aged 18–30 | 102 | 2.0 | 123 | 2.5 | 289 | 5.8 | 943 | 19.0 | 1375 | 27.7 | 2130 | 42.9 | 4.05 | 1.05 |
| People aged under 18 | 282 | 5.7 | 148 | 3.0 | 253 | 5.1 | 657 | 13.3 | 831 | 16.8 | 2788 | 56.2 | 4.25 | 1.08 |

Base: all participants (weighted). List order was randomised for each participant.
*Missing cases range from n=11 to n=21.
†Excludes 'should not be offered', missing cases range from n=45 to n=301.
SD, standard deviation.

trust in each. A majority (61%) indicated they had very little/no trust in the media (eg, newspapers/magazines/television/radio).

Trust did not differ by gender except for drug companies and the World Health Organisation (WHO), with females more likely to indicate trust in these sources (online supplemental tables S5 and S9, respectively).

Trust was higher among older participants for five sources (doctors/nurses/other healthcare professionals, NHS, UK government, media and family/friends; online supplemental tables S2, S4, S6, S10, S13). For example, trust in the UK government was higher among those aged 50–59 than 18–49 years (online supplemental table S6).

Trust varied by education. Compared with those without qualifications, other participants were more likely to trust five sources (doctors/nurses/other healthcare professionals, NHS, scientists, WHO; online supplemental tables

S2, S4, S8, S9) and less likely to trust another five (drug companies, media, social media, celebrities/social media influencers, family/friends; online supplemental tables S5, S10–S13). Compared with those 'living comfortably' participants in more difficult financial situations were less likely to trust the seven sources most closely aligned with scientific or clinical expertise (doctors/nurses/other healthcare professionals, pharmacists, NHS, drug companies, UK government, scientists, WHO; online supplemental tables S2-S6, S8, S9). Similarly, participants from minority ethnic groups were less likely to trust scientific or clinical sources than white British participants (online supplemental tables S2–S4, S8, S9). While lack of trust in faith or community leaders was low overall, Asian/Asian British participants were more likely than white British to trust faith/community leaders (AOR=4.82, 95% CI (2.76 to 8.42), p<0.001) as were black/black British participants

(AOR=4.52, 95% CI (2.04 to 9.99), p<0.001) (online supplemental table S14).

## Views on prioritisation

Nine in 10 participants rated healthcare professionals as highest priority for vaccination. Over 70% indicated those with serious health conditions/heightened vulnerability to COVID-19, care home workers and residents, and over 80s should be 'one of the first' to be vaccinated (table 4). Priority was also given to social care workers, school-teachers and those directly working with the public. Over one-third considered each of these groups should be 'one of the first' to be vaccinated, and 70% or more rated them in the top two priority levels. People aged under 18 were rated as lowest priority, and 6% considered the vaccine should not be offered to this group.

## Importance of second dose

Nearly all participants (96%, n=4761) considered it 'very' or 'fairly important' to receive the second vaccine dose. This increased to 99% (n=4096) among those who intended to accept the vaccine.

## DISCUSSION
### Principal findings

Overall, acceptance was high, with 83% having received or intending to have the vaccine. Acceptance increased with age and education, and if invited for vaccination. It decreased with financial hardship, and among non-white British ethnicities and those with unconfirmed past COVID-19. Clinical and scientific information was most trusted, with sociodemographic differences for different sources. Policy on a second dose and vaccination priority groups[1] was supported.

### Comparison with other studies

We confirmed lower acceptance in younger groups[6–8 10 11]; acceptance was higher if invited for vaccination, a finding observed for other vaccines in other populations,[23] and emphasising the importance of ensuring vaccine invitations are issued, using appropriate language with translations if necessary. Confirmation of lower acceptance in non-white British ethnicities[5 6 9 24] is concerning given increased risk of infection and poorer outcomes.[25] This lower acceptance has been reported to result from an erosion of trust with healthcare services as a consequence of past experiences of unethical experimental research conducted among black populations, the lack of participants from ethnic minorities included in health research, particularly vaccine trials, and poor experiences of healthcare.[15] Successful initiatives by primary care health professionals to overcome these barriers have been reported, but they require considerable resources.[26] We confirmed lower acceptance in those with lower educational attainment and greater financial hardship,[6 8–10 12 27] leaving these groups at risk of infection and increasing likelihood of emergence of variants.[28] Gender was not associated

with vaccine hesitancy in the analysis reported in this paper, but female gender has been found to be a factor associated with greater COVID-19 vaccine hesitancy in some other studies[6 8–10 29]; further research is needed to explore whether and why gender may relate to hesitancy.

A novel finding was that there was lower vaccine acceptance among those with unconfirmed but suspected COVID-19. This suggests that prior infection is thought to confer immunity, or that recovery fosters a perception of decreased severity, but further research is needed to explore this relationship. However, past infection does not guarantee protection and people may still be infectious.[30 31] Messaging should target those with prior infection.

There are other implications for communications. While high acceptance suggests communications are effective, identifying barriers in hesitant groups is a priority for developing interventions.[3 15 19 32] Trusted information sources are needed. The most trusted were the NHS, healthcare professionals, and scientific and medical advisers. This suggests that healthcare professionals have a central role in promoting vaccination in initiatives and during consultations. That government and media are less trusted has implications for acceptance.[7 8 27 33] We found particularly low levels of trust in social media and celebrities. However, this does not necessarily mean that they do not influence feelings about vaccination, and, with careful research, they could still play a positive role in communications (eg, initiatives using ethnic minority celebrities and opinion leaders).[16] Such initiatives would need to use pretesting of messages to ensure they are appropriately tailored to target audiences, while avoiding stereotyping, and would require evaluation of acceptability and effectiveness.

Differences in trust varied by sociodemographics. Compared with white British participants, other ethnicities had lower trust in healthcare and scientific sources. Although trust in faith/community leaders was low, it was higher in Asian and black British participants, suggesting a role for these leaders.[15] Those with lower educational attainment or financial hardship had lower trust in healthcare and scientific sources. Those with no qualifications had higher trust in media and family/friends. This suggests a need for a mix of information sources for these groups. Mainstream media may have a role to play, despite lower trust.[27]

Reassuringly for further campaigns, for the first time, this study reported that prioritisation was considered acceptable by the general public and there was support for additional prioritisation of schoolteachers and others in direct contact with the public. This is consistent with research suggesting that healthcare workers themselves support the decision to prioritise vaccination for front-line health and social care workers and those at increased risk of vulnerability to infection.[34] As planning begins for further vaccination, careful communication regarding prioritisation should continue. We found high support for a second dose, suggesting the UK's decision to extend the

period between doses has not dented public confidence. While the high acceptance rate may suggest that acceptance will be similarly high in future COVID-19 vaccination programmes, this cannot be assumed. The survey was conducted during a period of considerable public anxiety, with rising infection rates and restrictions on many activities including travel. Similar acceptance rates may not be observed in future if the threat is perceived to have receded and society is functioning more normally.

## Strengths and limitations

Strengths include the large probability-based nationally representative sample, ability to analyse by ethnicity and surveying during vaccine roll-out. Our findings can be generalised to GB's adult population, however global contexts for COVID-19 and vaccination vary. Although not generalisable to them, the findings are still informative for other countries. The study has limitations. As it is cross-sectional, we cannot infer causality; although we included variables likely to be important in vaccine acceptance, these results are exploratory. Our qualitative studies will deepen understanding of associations. A survey repeated when COVID-19 cases and deaths are low, and without lockdown, might yield different responses. We did not survey individuals who are institutionalised (eg, prisoners), notably difficult to reach (eg, homeless) or those not speaking English (therefore, our ethnic minority sample may under-represent certain views); specific surveys are needed for these groups. We investigated vaccination intention. Actual uptake may be lower, although it is likely that factors associated with intention will influence uptake.

## CONCLUSIONS

COVID-19 vaccination acceptance is high in GB. Targeted engagement is needed to address hesitancy in non-white British ethnic groups, those with lower education, those younger, those with greater financial hardship and those with unconfirmed but suspected past infection. Healthcare professionals and scientific advisors should lead communications and tailoring is needed. Work is needed to rebuild trust in government information. There is high support for having the second vaccine dose. Views of vaccine prioritisation are mostly consistent with UK official policy but there was support for prioritising additional groups and careful communication around vaccination prioritisation should continue.

**Acknowledgements** We thank the questionnaire development and testing and survey and data delivery teams at NatCen for their work on the survey, and Professor Mark Petticrew at the London School of Hygiene & Tropical Medicine for acting as adviser to the study.

**Contributors** MS, CJ, HB, KH and AMM conceived the study, supported by AF, DE and AMM. MS, CJ, KA, HB and AMM designed the questionnaire, supported by MU, AF, DE, AM and KH. CJ and AMM acquired and analysed the data, which was interpreted by MS, CJ, HB, MU, KH and AMM. MS and AMM drafted the manuscript supported by CJ, HB, MU and KH. KA, HB, MU and KH critically revised the article, supported by MS, CJ, AF, DE, AM and AMM. All authors read the final version of the manuscript and gave approval for it to be published. AMM, CJ and MS had access

to the data in the study and take responsibility for the integrity of the data and the accuracy of the data analysis. The corresponding author attests that all listed authors meet authorship criteria and that no others meeting the criteria have been omitted. MS is the guarantor.

**Funding** The OPTIMising general public Uptake of a COVID-19 vaccine (OPTIMUM) study was supported by a UK Research & Innovation (UKRI) Ideas to Address COVID-19 award (no. ES/V012851/1).

**Competing interests** KH has received another UK Research and Innovation (Economic and Social Research Council) grant on the impact of COVID-19.

**Patient consent for publication** Not applicable.

**Ethics approval** The study received ethical approval from NatCen's Research Ethics Committee (ID P14307). Participants gave informed consent before taking part.

**Provenance and peer review** Not commissioned; externally peer reviewed.

**Data availability statement** All data relevant to the study are included in the article or uploaded as online supplemental information. After completion of the study, the survey dataset will be deposited in the UK Data Archive.

**ORCID iDs**
Kathryn Angus http://orcid.org/0000-0002-5351-4422
Kate Hunt http://orcid.org/0000-0002-5873-3632

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
