## [Reviewer comments · BMJ Open]

ARTICLE DETAILS

TITLE (PROVISIONAL)	A national survey of attitudes towards and intentions to vaccinate against COVID-19: implications for communications
AUTHORS	Stead, Martine; Jessop, Curtis; Angus, Kathryn; Bedford, Helen; Ussher, Michael; Ford, Allison; Eadie, Douglas; MacGregor, Andy; Hunt, Kate; MacKintosh, Anne Marie

VERSION 1 – REVIEW

REVIEWER	Petravić, Luka University of Maribor
REVIEW RETURNED	20-Jul-2021

GENERAL COMMENTS	I would like to congratulate the authors on a finely written manuscript. Manuscript “A national survey of attitudes towards and intentions to vaccinate against COVID-19: implications for communications” by Stead et al. explores vaccine hesitancy in the UK. The manuscript is inside the scope of the journal. The study took place in early 2021 (January to February). The study aimed to examine public views on the COVID-19 vaccination and consider the implications for communications. The research question is clearly outlined and justified given the existing body of evidence. The study results in the abstract are written clearly. The title is informative and relevant. The references are relevant, recent and have appropriate key studies included. The introduction outlines what is already known about this topic. The variables are defined and measured appropriately. The study design was appropriate to answer the aim of the study. The study uses randomly selected participants that are part of the NatCen panel. STROBE statement was used to better present the study and make it more replicable. Because of the use of the panel and data set weighing the results should be closer to reality than in research using lesser methods of sampling. The sample is also quite large for a panel survey, which makes it possible to create better conclusions. The timing of the survey also made it so that the option of the vaccine is not hypothetical but existent. All of the data is presented in the results. Text in results adds to data as it highlights significant findings. The discussion is solidly written and encompasses all of the important findings. Authors have tried to correlate demographic attributes to the intention to get vaccinated. They have also looked closer at the
---

	attitudes to vaccine prioritisation, trust in information sources and importance of the second dose. They find that there are significant differences when comparing younger individuals to older, comparing races, education, self-reported income, offering vaccination on an individual level and COVID-19 status. The study limitations are good opportunities to inform further research in future. Limitations should include that for other than white ethnicity the sample was quite small (a few thousand versus a few hundred), which can skew the results from the genuine state. I could find no major flaws in this article. The article is consistent within itself. There are however minor details that could be addressed in my opinion to make this manuscript better.  1. Reference 7 has been corrected after the citation and has a changed doi, this should be updated. 2. Reference 13 has been updated since the citation, it is necessary to include the access date or update the reference. 3. Reference 23 is a pre-print and has not been, to the best of my knowledge, peer-reviewed yet. Nothing can be done here. 4. Introduction: The introduction should put the UK results into a European context – who has higher vaccine acceptance? 5. Methods: “Small financial sum” should be defined. 6. Supplement: A questionnaire should be added as a supplement in a non-paragraph form, to make it easier for the reader to see and understand the questions posed. 7. Supplement: The data-set used to weigh the current data-set should be in supplement/depository, to better show how the demographics from the survey differ from those of ONS. 8. Methods: How did you choose who got called and who answered online? 9. Methods: A statistically significant result should be defined in methods under data analysis, in such big datasets a $p < 0,01$ could be warranted. 10. Page 6, line 14: How was the test group chosen (first 20 participants), what were its characteristics? 11. Methods: STROBE item 10: there is no description specifically saying how did you come to the current number of participants. Was this the whole panel group or did you use a smaller subgroup? 12. Results: Tables 3 and 4 could be presented with a figure also, a stacked bar chart would be a nice representation. 13. Discussion: More added value could be added to the discussion by more thoroughly comparing the current body of evidence with your results. A lot of research comments on the results in respect of gender, this could be further explored in the current manuscript; this could be done by comparing vaccine hesitancy between the genders and then comparing outcomes with scientific literature, as there can be a quite big difference between females and males. 14. Page 9, line 13: Why would it alter, is there any article published describing this? 15. Page 9, line 37: is this in line with other surveys or is this first described in this manuscript? 16. Page 9, line 51: what would this sophisticated tailoring entail? 17. Discussion: What could be the reasons for differences between races and their vaccine hesitancy?
--	---

	18. Discussion: Was prioritisation researched first in this paper, could you find an article where this was already researched and compare it to your outcomes? 19. Discussion: The decision of the UK to delay 2nd dose has not dented public confidence, was maybe this decision in other countries met with more resistance or decline in vaccine acceptance? 20. Tables throughout: When comparing tables in manuscript and tables in supplement percents are written differently (example 87 % and 87), this should be unified (either you write “%” in each cell or you put % in the header row).
--	--

REVIEWER	Lauri, Josef University of Malta, Mathematics
REVIEW RETURNED	31-Jul-2021

GENERAL COMMENTS	A very comprehensive study and excellently presented. While a complete understanding of the statistical analysis would require some specialist knowledge, the findings and conclusions are very clearly explained and can be understood by most readers, from the interested member of the general public to anyone involved in some way in tackling the COVID-19 problem. One looks forward to the qualitative study which the authors have announced they will be carrying out.
---

REVIEWER	Islam, Md. Saiful Jahangirnagar University, Public Health and Informatics
REVIEW RETURNED	03-Aug-2021

GENERAL COMMENTS	 1. In the introduction, the authors can include the current status of COVID-19 vaccination of the UK population. 2. The last paragraph of the introduction would be more appropriate to place in the methods sections. 3. How the sample was drawn or selected? Which probability technique? 4. What are the inclusion and exclusion criteria? 5. The findings are not very surprising. 6. What are the public health implications of the present findings.
--

VERSION 1 – AUTHOR RESPONSE

Reviewer: 1
Dr. Luka Petravić, University of Maribor

Comments to the Author:

I would like to congratulate the authors on a finely written manuscript.

Manuscript “A national survey of attitudes towards and intentions to vaccinate against COVID-19: implications for communications” by Stead et al. explores vaccine hesitancy in the UK. The manuscript is inside the scope of the journal. The study took place in early 2021 (January to

February). The study aimed to examine public views on the COVID-19 vaccination and consider the implications for communications. The research question is clearly outlined and justified given the existing body of evidence. The study results in the abstract are written clearly. The title is informative and relevant. The references are relevant, recent and have appropriate key studies included. The introduction outlines what is already known about this topic.

The variables are defined and measured appropriately. The study design was appropriate to answer the aim of the study. The study uses randomly selected participants that are part of the NatCen panel. STROBE statement was used to better present the study and make it more replicable. Because of the use of the panel and data set weighing the results should be closer to reality than in research using lesser methods of sampling. The sample is also quite large for a panel survey, which makes it possible to create better conclusions. The timing of the survey also made it so that the option of the vaccine is not hypothetical but existent.

All of the data is presented in the results. Text in results adds to data as it highlights significant findings.

The discussion is solidly written and encompasses all of the important findings.

Authors have tried to correlate demographic attributes to the intention to get vaccinated. They have also looked closer at the attitudes to vaccine prioritisation, trust in information sources and importance of the second dose. They find that there are significant differences when comparing younger individuals to older, comparing races, education, self-reported income, offering vaccination on an individual level and COVID-19 status.

The study limitations are good opportunities to inform further research in future.

Limitations should include that for other than white ethnicity the sample was quite small (a few thousand versus a few hundred), which can skew the results from the genuine state.	We have revised the Strengths and Limitations section as requested (see above). However, we have not included this item as a limitation. The smaller number of participants from non-white ethnicity is to be expected given their lower prevalence in the population. Descriptive data have been weighted to reflect the demographic profile of adults in Great Britain. All the multivariate analyses control for key demographic characteristics, including ethnicity.
I could find no major flaws in this article. The article is consistent within itself.	Thank you for this feedback.

There are however minor details that could be addressed in my opinion to make this manuscript better.	
1. Reference 7 has been corrected after the citation and has a changed doi, this should be updated.	Thank you for alerting us to this. Reference 7 (Lazarus et al. 2021) has a published Author Correction (with its own doi https://doi.org/10.1038/s41591-020-01226-0) and both the html and PDF versions of Reference 7 have been corrected. As far as we can see, the doi for the corrected article/Reference 7 remains unchanged (https://doi.org/10.1038/s41591-020-1124-9). We have checked Lazarus et al.'s corrections and are satisfied that we do not have to change how we have used their findings. However, we should have cited the full paper, instead of its supplementary file only; this has now been corrected. (We cited findings from the supplementary information in our Introduction and from both the supplementary information and the full paper in our Discussion.)
2. Reference 13 has been updated since the citation, it is necessary to include the access date or update the reference.	We have added the accessed date to Reference 13.
3. Reference 23 is a pre-print and has not been, to the best of my knowledge, peer-reviewed yet. Nothing can be done here.	To date, this study remains a pre-print.
4. Introduction: The introduction should put the UK results into a European context – who has higher vaccine acceptance?	We discussed this comment, and comment #1 by Reviewer 3, carefully. Although we understand the reviewers' preferences for the manuscript to refer to UK vaccine uptake and to compare this with uptake in other European countries, we are not convinced that this would be particularly meaningful for the reader. The UK COVID-19 vaccination rate changes every day, and therefore we would need to decide which date to use (this dilemma would also apply to the inclusion of any uptake data from other countries). If we include the current vaccination uptake (ie. today's date), this will not reflect the uptake rate earlier in the year, when the study was conducted, and will be out of date by the time the article is published (if accepted). We could attempt to include the uptake rate when the survey was conducted, as this is the context in which participants provided their responses. However, as the survey was conducted over

	several weeks, it is not straightforward to identify the appropriate figure, as the uptake would have increased daily over the survey period. Earlier responders would have completed the survey when the vaccination uptake rate was lower than for later responders. On balance, therefore, we feel that it would not be particularly meaningful to describe the vaccination uptake on an arbitrary date.
5. Methods: "Small financial sum" should be defined.	Participants receive between £5 and £20 depending on a number of factors. We have now added this information in the manuscript. New text reads: "(£5 - £20 depending on interview duration and whether participant had characteristics which are typically under-represented in survey samples)" (page 5).
6. Supplement: A questionnaire should be added as a supplement in a non-paragraph form, to make it easier for the reader to see and understand the questions posed.	We have addressed this and provided a new supplement containing the questionnaire. See Supplementary Material, Methods S1.
7. Supplement: The data-set used to weigh the current data-set should be in supplement/depository, to better show how the demographics from the survey differ from those of ONS.	We have provided a new supplementary file explaining the weighting approach, and variables used, including links to additional documentation on the recruitment and how the data used in the weighting were collected. See Supplementary Material, Methods S2.
8. Methods: How did you choose who got called and who answered online?	We have expanded the text here to explain which respondents were invited to take part, and to explain which participants answered online or by telephone. The text now reads: "All BSA respondents who agreed to join the Panel, had not requested to leave or become inactive were invited to take part, maintaining the random probability design. Data were collected through online and telephone interviews (conducted 14th January to 7th February 2021). Panellists were sent reminders and offered a small financial sum (£5 - £20 depending on interview duration and whether participant had characteristics which are typically under-represented in survey samples) in recognition of their contribution. Participants who did not initially take part online, and for whom a telephone number was available, were followed up by a telephone interviewer and encouraged to take part online or given the opportunity to take part on the telephone." (page 5).
9. Methods: A statistically significant result should be defined in methods under data	The following sentence has been added to the data analysis section: "Given the large sample

analysis, in such big datasets a $p < 0,01$ could be warranted.	size in this study, the threshold for statistical significance was set at $p < 0.01$.” (page 7).
10. Page 6, line 14: How was the test group chosen (first 20 participants), what were its characteristics?	We have expanded the text here to provide further information on the recruitment and selection of the test group. The text now reads: “Interviews were conducted with 20 individuals recruited by an external fieldwork agency. A purposive sampling approach was employed, with quotas used to ensure people with a mix of genders, ages, parental status, likelihood of accepting a COVID-19 vaccination, and experiences of shielding were recruited.” (page 5). We felt it was not necessary to include detailed characteristics of the test group in the manuscript, but for information they comprised: 10 male, 10 female 4 18-34; 5 35-49; 4 50-69; 7 70+ 9 parents of children aged 18 or under (11 not) 14 likely to get a vaccine, 5 unlikely, 1 DK 8 had experienced shielding due to being higher risk, 12 had not
11. Methods: STROBE item 10: there is no description specifically saying how did you come to the current number of participants. Was this the whole panel group or did you use a smaller subgroup	We have now added this information in the manuscript (see response to comment #8 above), and the STROBE checklist directs the reader to this section of the manuscript.
12. Results: Tables 3 and 4 could be presented with a figure also, a stacked bar chart would be a nice representation.	We feel that the tables are sufficient as they present all the necessary information to the reader. However, if the editor would prefer figures or bar charts, we are happy to prepare these.
13. Discussion: More added value could be added to the discussion by more thoroughly comparing the current body of evidence with your results. A lot of research comments on the results in respect of gender, this could be further explored in the current manuscript; this could be done by comparing vaccine hesitancy between the genders and then comparing outcomes with scientific literature, as there can	We thank the reviewer for raising this, because it prompted us to think carefully about this point. Gender was not a factor in the analysis reported in this paper, but it has been found to be a factor in some other studies, and we recognise that gender may emerge as important in further analysis of our data when we take into account other potential determinants of acceptance, such as attitudes. We have therefore amended the text so that it now reads: “Gender was not associated with vaccine hesitancy in the analysis

be a quite big difference between females and males.	reported in this paper, but female gender has been found to be a factor associated with greater COVID-19 vaccine hesitancy in some other studies;^{6,8-10,29} further research is needed to explore whether and why gender may relate to hesitancy.” (page 10) and added a new citation: 29 Malik AA, McFadden SM, Elharake J, Omer SB. Determinants of COVID-19 vaccine acceptance in the US. EClinicalMedicine. 2020;26:100495. https://doi.org/10.1016/j.eclinm.2020.100495
14. Page 9, line 13: Why would it alter, is there any article published describing this?	Thank you for this comment – it encouraged us to reconsider the statement we had made in the manuscript about the implications for future vaccine acceptance. On reflection, the important point here is that the survey was conducted at a point in time when the COVID-19 situation in the UK was perceived to be severe, and it cannot be assumed that there would be similarly high acceptance of future vaccination. We have amended the text so that it now reads: “While the high acceptance rate may suggest that acceptance will be similarly high in future COVID-19 vaccination programmes, this cannot be assumed. The survey was conducted during a period of considerable public anxiety, with rising infection rates and restrictions on many activities including travel. Similar acceptance rates may not be observed in future if the threat is perceived to have receded and society is functioning more normally” (page 11). We have also moved this text to later in the Discussion section, as it relates more to implications for the future.
15. Page 9, line 37: is this in line with other surveys or is this first described in this manuscript?	We are not aware of this having been widely reported previously. We have amended the text to frame the finding as potentially novel, and have also flagged up the need for future research to explore further why prior suspected infection is associated with lower acceptance. The text now reads: “A novel finding was that there was lower vaccine acceptance among those with unconfirmed but suspected COVID-19. This suggests that prior infection is thought to confer immunity, or that recovery fosters a perception of decreased severity, but further research is needed to explore this relationship.” (page 10).

16. Page 9, line 51: what would this sophisticated tailoring entail?	We agree that this sentence could be clearer and have rewritten to clarify: “We found particularly low levels of trust in social media and celebrities. However, this does not necessarily mean that they do not influence feelings about vaccination, and, with careful research, they could still play a positive role in communications (for example, initiatives using ethnic minority celebrities and opinion leaders show promise.¹⁶) Such initiatives would need to use pre-testing of messages to ensure they are appropriately tailored to target audiences, while avoiding stereotyping, and would require evaluation of acceptability and effectiveness.” (page 10).
17. Discussion: What could be the reasons for differences between races and their vaccine hesitancy?	We have added new text setting out potential reasons for vaccine hesitancy in minority ethnic populations: “This lower acceptance has been reported to result from an erosion of trust with health care services as a consequence of past experiences of unethical experimental research conducted among black populations, the lack of participants from ethnic minorities included in health research, particularly vaccine trials, and poor experiences of healthcare.¹⁵ Successful initiatives by primary care health professionals to overcome these barriers have been reported, but they require considerable resources.²⁶” (page 9) and cited another paper: 26 Mahase E. Covid-19 vaccines: GPs boost uptake by calling patients and teaming up with community groups BMJ 2021;374:n2093.
18. Discussion: Was prioritisation researched first in this paper, could you find an article where this was already researched and compare it to your outcomes?	As far as we are aware this is the first study to research prioritisation in the general UK public. We are aware of a study which has examined healthcare workers’ views of vaccine implementation, including their views on prioritisation, and have added a line in the Discussion: “This is consistent with research suggesting that healthcare workers themselves support the decision to prioritise vaccination for frontline health and social care workers and those at increased risk of vulnerability to infection.³⁴” (page 10). We have added it to the reference list: 34 Manby R, Dowrick A, Karia A, Maio L, Buck C, Singleton G, et al. Healthcare workers’ perceptions and attitudes towards the UK’s COVID-19 vaccination programme. medRxiv 2021; published online Mar 31. https://doi.org/10.1101/2021.03.30.21254459 (preprint).

19. Discussion: The decision of the UK to delay 2nd dose has not dented public confidence, was maybe this decision in other countries met with more resistance or decline in vaccine acceptance?	We have not managed to identify an academic study that measures attitudes to second dose uptake in other countries who have delayed provision; unless the reviewer is aware of one that they are able to share.
20. Tables throughout: When comparing tables in manuscript and tables in supplement percents are written differently (example 87 % and 87), this should be unified (either you write “%” in each cell or you put % in the header row).	We have made the tables consistent now by putting the % in the header row in all tables.

Reviewer: 2

Dr. Josef Lauri, University of Malta

Comments to the Author:

A very comprehensive study and excellently presented. While a complete understanding of the statistical analysis would require some specialist knowledge, the findings and conclusions are very clearly explained and can be understood by most readers, from the interested member of the general public to anyone involved in some way in tackling the COVID-19 problem. One looks forward to the qualitative study which the authors have announced they will be carrying out.

We thank you for this feedback.

Reviewer: 3

Mr. Md. Saiful Islam, Jahangirnagar University

Comments to the Author:

1 In the introduction, the authors can include the current status of COVID-19 vaccination of the UK population	We discussed this comment, and comment #4 by Reviewer 1, carefully. Although we understand the reviewers' preferences for the manuscript to refer to UK vaccine uptake (and, for Reviewer 1, to compare this with uptake in other European countries), we are not convinced that this would be particularly meaningful for the reader. The UK COVID-19 vaccination rate changes every day, and therefore we would need to decide which date to use (this dilemma would also apply to the inclusion of any uptake data from other countries). If we include the current vaccination uptake (ie. today's date), this will not reflect the uptake rate earlier in the year, when the study was conducted, and will be out of date by the time the article is published (if accepted). We could attempt to include the uptake rate when the survey was conducted, as this is the context
--	--

	in which participants provided their responses. However, as the survey was conducted over several weeks, it is not straightforward to identify the appropriate figure, as the uptake would have increased daily over the survey period. Earlier responders would have completed the survey when the vaccination uptake rate was lower than for later responders. On balance, therefore, we feel that it would not be particularly meaningful to describe the vaccination uptake on an arbitrary date.
2 The last paragraph of the introduction would be more appropriate to place in the methods sections	We feel that this paragraph is setting out the aim of the study and explaining the context in which the survey was conducted (ie. when the vaccination programme had begun, targeting older adults first), and therefore it works better at the end of the Introduction. However, if the editor advises that we should move it to the Methods section, we are happy to do so.
3 How the sample was drawn or selected? Which probability technique?	We have expanded the text here to explain which respondents were invited to take part, and to explain which participants answered online or by telephone. The text now reads: "All BSA respondents who agreed to join the Panel, had not requested to leave or become inactive were invited to take part, maintaining the random probability design. Data were collected through online and telephone interviews (conducted 14th January to 7th February 2021). Panellists were sent reminders and offered a small financial sum (£5 - £20 depending on interview duration and whether participant had characteristics which are typically under-represented in survey samples) in recognition of their contribution. Participants who did not initially take part online, and for whom a telephone number was available, were followed up by a telephone interviewer and encouraged to take part online or given the opportunity to take part on the telephone." (page 5).
4 What are the inclusion and exclusion criteria?	We have added a sentence to clarify that: "The target population for the study was adults (18+) living in Great Britain." (page 5).
5 The findings are not very surprising	No action needed.
6 What are the public health implications of the present findings	We feel that the public health implications of the findings are already presented throughout the Discussion. We highlight the following discussion of these implications:

	 • “...the importance of ensuring vaccine invitations are issued, using appropriate language with translations if necessary.” (page 9) • That lower acceptance in non-White British ethnicities “... is concerning given increased risk of infection and poorer outcomes.” (page 9) • “Messaging should target those with prior infection.” (page 10) • “...identifying barriers in hesitant groups is a priority for developing interventions.” (page 10) • “Trusted information sources are needed. The most trusted were the NHS, healthcare professionals, and scientific and medical advisers. This suggests that healthcare professionals have a central role in promoting vaccination in initiatives and during consultations. That government and media are less trusted has implications for acceptance.^{7,8,27,33} We found particularly low levels of trust in social media and celebrities. However, this does not necessarily mean that they do not influence feelings about vaccination, and, with careful research, they could still play a positive role in communications (for example, initiatives using ethnic minority celebrities and opinion leaders.¹⁶)” (page 10) • “Although trust in faith/community leaders was low, it was higher in Asian and Black British participants, suggesting a role for these leaders.¹⁵ Those with lower educational attainment or financial hardship had lower trust in healthcare and scientific sources. Those with no qualifications had higher trust in media and family/friends. This suggests a need for a mix of sources for these groups. Mainstream media may have a role to play, despite lower trust.²⁷” (page 10) • “Reassuringly for further campaigns, for the first time, this study reported that prioritisation was considered acceptable by the general public and there was support for additional prioritisation of schoolteachers and others in direct contact with the public.” (page 10) • “We found high support for a second dose, suggesting the UK’s decision to extend the period between doses has not dented public confidence.” (page 10)
--	---